

# Atmospheric Forcing as a driver for Ocean Forecasting

Andreas Schiller[1], Simon A. Josey[2], John Siddorn[2], Ibrahim Hoteit[3]

[1]CSIRO Environment, Castray Esplanade, Hobart, Tasmania, Australia
[2]National Oceanography Center, Southampton, UK
[3]Physical Science and Engineering Division, King Abdullah University of Science and Technology (KAUST), Thuwal, Saudi Arabia

*Correspondence to*: Andreas Schiller (anschill2020@gmail.com)

**Abstract.** The connection of the ocean component with the Earth system is subject to the way the atmosphere interacts with it. The paper illustrates the state of the art in the way atmospheric fields are used in ocean models as boundary conditions for the provisioning of the exchanges of heat, freshwater and momentum fluxes. Such fluxes can be based on remote-sensing instruments, like SAR, or provided directly by Numerical Weather Prediction systems. This study also discusses how the ocean-atmosphere fluxes are numerically ingested in ocean models from global to regional to coastal scales. Today's research frontiers on this topic are opening challenging opportunities for developing more sophisticated coupled ocean-atmosphere systems.

## 1 Air-Sea Flux Data Sets

The exchanges of heat, freshwater and momentum between the oceans and the atmosphere play a critical role as boundary conditions in global, regional and coastal Operational Ocean Forecasting Systems (OOFS). A brief overview, including uncertainties, of air-sea flux data sets of heat, freshwater and momentum (which is equivalent to wind stress) is presented in Section 2 with applications in OOFS in mind.

Nowadays, the two primary sources of information regarding air-sea fluxes used in OOFS are satellite observations and atmospheric model forecasts which assimilate various data types. In recent years, several new flux products have become available which contain fields at sub-daily and hourly timescales. This tendency has been driven, in part, by the high time resolution possible with atmospheric forecasts and the need to include high frequency variability in forcing fields for OOFS. A complete survey of the wide range of flux datasets and their technical details is beyond the scope of this document. Instead, an overview of the main flux datasets is presented in Section 2 with the most used data sets in OOFS being highlighted.

For information about sea-ice boundary conditions we refer the reader to Section 2, noting that sea-ice models can be part of an OOFS, of a Numerical Weather Prediction (NWP) system or be coupled to both. Consequently, respective input sourced from external data sets depend on the exact model architecture.



## 1.1 Remotely-sensed fluxes

All OOFS known to us today use air-sea fluxes provided by NWP systems, mostly because of low latency required for OOFS and the convenience of NWP outputs being gridded products. These NWP systems often assimilate relevant satellite observations to obtain improved estimates of air-sea fluxes, hence we briefly describe some of the remotely sensed observations in the subsequent paragraphs. Furthermore, remotely sensed estimates of air-sea fluxes can be used to validate the surface fluxes in an OOFS.

The net air-sea heat flux is the sum of four components: two turbulent heat flux terms (the latent and sensible heat fluxes) and two radiative terms (the shortwave and longwave fluxes). Bulk formulae are employed to estimate the latent and sensible heat fluxes whereas radiative fluxes are determined either from empirical formulae or from radiative transfer models (Josey, 2011). Satellite-based estimates of air-sea heat flux terms suffer because it is not yet possible to reliably measure near surface air temperature and humidity directly from space. For example, satellites measure radiances in various wavelength bands which must then be inverted to obtain temperature. These indirect techniques lead to a source of uncertainty in the turbulent heat flux terms which are critically dependent on the sea-air temperature and humidity difference near the interface (Hooker et al., 2018; Tomita et al., 2018). Estimates of the radiative flux terms are available from various sources, e.g. Pinker et al. (2018), and can be combined with indirect estimates of the turbulent fluxes to form net heat flux products.

In contrast, the wind stress is well determined since the launch of QuikSCAT in 1999 (Hoffman and Leidner, 2005) and subsequent satellite missions. Global wind measurements by Synthetic Aperture Radar (SAR) go all the way up to the coast due to its high resolution, filling critical gaps in ocean wind speed and direction observations in coastal areas (Khan et al., 2023).

Precipitation is also remotely-sensed using various techniques including infrared measurements of cloud top brightness temperature, which acts as a proxy for rain rate, and passive microwave measurements. Launched in 2014, the US-Japanese led Global Precipitation Measurement Mission (GPM) is an international network of satellites that provides global observations of rain and snow at different times of the day (Hou et al., 2014). However, validation of these fields over the ocean is challenging due to the lack of high quality measurements from rain sensors and the difficulty with making this measurement (Weller et al., 2008). As a consequence, uncertainty remains in the precipitation fields with follow-on effects for estimating the associated air-sea freshwater flux (evaporation minus precipitation) (Josey, 2011).

## 1.2 Numerical weather prediction

NWP models assimilate a wide range of observations including surface meteorological reports, radiosonde profiles and remote sensing measurements. These models have become a major source of providing to OOFS complete sets of air-sea flux fields at high resolution (3 hourly or better) with global spatial coverage. The turbulent flux terms are estimated from the model surface meteorology fields while the shortwave and longwave flux are output from the radiative transfer component of the atmospheric model. Air-sea fluxes from NWP systems are an attractive option for OOFS because of their operational reliability



and timely release of forcing fields akin to the operational cycles of OOFS. However, NWP systems are of course dependent
on the model physics which, although constrained to some extent by the assimilated observations, has the potential to produce
biases, particularly in the radiative flux fields and precipitation (Trenberth et al., 2009; Weller et al., 2022).

Fixed versions of NWP models run over multidecadal periods are commonly referred to as atmospheric reanalyses - two
examples being those from the National Center for Environmental Prediction and the National Center for Atmospheric
Research (NCEP/NCAR) and ECMWF. Although not suitable for near real-time OOFS due to their delayed-mode operation,
air-sea fluxes derived from atmospheric reanalyses have proven to be a valuable tool for testing OOFS during their
development stages as well as scenario simulations and analyses of past extreme events. Table 4.4-1 provides further examples
of widely used global atmospheric NWP and some atmospheric reanalysis products.

### 1.3 Other flux products

In addition to the two primary classes of flux datasets described above, flux fields for OOFS are available from several other
types of products. An example are surface fluxes available from various ocean synthesis efforts, that is ocean models with data
assimilation such as the Estimating the Circulation and Climate of the Ocean (ECCO) model (Stammer et al., 2004). These are
typically forced by global atmospheric reanalysis fields which are then adjusted as a result of the assimilation and optimisation
process. Similar to atmospheric reanalyses, air-sea datasets based on delayed-mode synthesis efforts are suitable for testing
OOFS during their development stages.

### 2 Applications of Air-Sea Flux Data Sets in OOFS

Typically, NWP systems produced by national meteorological services provide atmospheric surface forcing fields to OOFS in
order to compute water, heat, and momentum fluxes. Such fields may be also supplemented by real-time or near real-time
observations, e.g. satellite data, and other averaged datasets including climatology. Alternatively, in a more complex modelling
framework, an ad hoc atmospheric model can be developed at the same resolution of the ocean model in order to provide high
resolution atmospheric fields (Alvarez-Fanjul et al., 2022).

Each of the classes of flux product described above has its own advantages and disadvantages and it is not possible to
recommend a best air-sea flux product; rather, the choice of flux dataset must be guided by the scientific feasibility and by the
application in mind. For example, near real-time NWP products are needed for operational ocean forecasting purposes whereas
a reanalysis product might be appropriate and more convenient to use during the development stages of an OOFS and for
validation purposes. We offer some examples of possible air-sea forcing fields in OOFS but are by no means complete or
prescriptive.



**Table 1: Examples of global atmospheric forcing products and providers. Adapted from Alvarez-Fanjul et al. (2022).**

| Dataset | Description | Provider |
|---|---|---|
| GFS | Global Forecast System, produced by the National Centers for Environmental Prediction (NCEP), provides analysis and forecast atmospheric fields for the global ocean at the resolution of about 28 km | https://www.ncdc.noaa.gov/data-access/model-data/model-datasets/global-forecast-system-gfs |
| NAVGEM | Navy Global Environmental Model runs by the United States Navy's Fleet Numerical Meteorology and Oceanography Center (FNMOC) | https://www.usno.navy.mil/FNMOC/meteorology-products-1m |
| ECMWF IFS and ERA5 | European Center for Medium range Weather Forecasting that provides reanalysis, analysis and forecast atmospheric fields at medium, extended, and long range | ECMWF https://www.ecmwf.int/ |
| Met Office UK | United Kingdom Meteorological Office that produces the Unified Model, a numerical model of the atmosphere used for both weather and climate applications | Met Office https://www.metoffice.gov.uk/ |
| GEM | Global Environmental Multiscale model, an integrated forecasting and data assimilation system developed in the Recherche en Prévision Numérique (RPN), Meteorological Research Branch (MRB), and the Canadian Meteorological Centre (CMC) | Environment Canada https://collaboration.cmc.ec.gc.ca/ |

## 2.1 Applications in Global OOFS

Global NWP models like those operated by centers listed in Table 1 at present have typical horizontal grid resolutions of 20 km or better (and 60 vertical levels or more). With this kind of horizontal resolution, it is possible to capture large-scale synoptic weather phenomena and associated signals in the air-sea fluxes used to force ocean models.

However, in NWP systems with such grid resolutions it is not possible to accurately simulate smaller-scale ocean-atmosphere interactions such as oceanic fronts, orographic features like land-sea circulation or air-sea interactions associated with
mesoscale oceanic eddies, noting that the synoptic (eddy-)scale in the ocean is of the order of ~100 km which is about an order of magnitude smaller than in the atmosphere at about ~1000 km.

Atmospheric forcing fields are typically interpolated onto the respective grid points of the ocean model, e.g. momentum fluxes onto the velocity grid points, air-sea heat fluxes onto the temperature grid points and evaporation minus precipitation onto the salinity grid points of the ocean model (plus volume or mass flux in the continuity equation). This interpolation can be
accomplished by either using an internal interpolation routine of the ocean model, by using bulk formulae at the ocean grid to



calculate surface fluxes of heat, freshwater and momentum or by using specific coupling software, e.g. Craig et al. (2017), for fully coupled ocean-atmosphere-wave-sea-ice models.

## 2.2 Applications in regional and coastal OOFS

There is a plethora of regional and coastal ocean models with fixed, variable and adaptive grids and with horizontal resolutions
often in the 10-100 m range (Kourafalou et al., 2015). It is therefore not possible to provide specific guidance about the appropriate choice of air-sea fluxes required for this type of models.

Regional to basin-scale OOFS are typically forced with air-sea-fluxes from the latest high-resolution global NWP systems, e.g. O'Dea et al. (2012). In contrast, coastal OOFS require a different approach. Coastal air-sea circulation and topographic features like small islands and their interactions with air-sea fluxes are not reproduced by global-scale atmospheric models,
hence much higher resolution coastal atmospheric models are needed to provide reliable upper ocean boundary conditions. This can be accomplished by direct coupling of high-resolution atmospheric models to coastal ocean models or by using air-sea fluxes from a stand-alone NWP higher resolution coastal model, e.g. Hordoir et al. (2019). Either way, these atmospheric models need to be (multiply) nested within coarser-resolution regional and/or global models which provide lateral and upper boundary conditions. This is an active field of R&D where the development of coastal NWP and OOFS often goes hand-in-
hand with efforts to develop fully coupled ocean-atmosphere forecasting systems. However, it should be noted that for both components, atmosphere and ocean, not just suitable lateral boundary conditions from coarser component models are required but it is also highly desirable to have an appropriately dense atmospheric and oceanic observing system to constrain these models and improve (coupled) forecasts.

High-resolution air-sea fluxes which are based on remotely sensed fluxes can also be used to evaluate the quality of the forcing
fields in coastal ocean models. An example is the Synthetic Aperture Radar (SAR)-based remotely sensed regional ocean wind speed and direction database which has been made available recently by the Australian Integrated Marine Observing System (Khan et al., 2023). The data set is a km-resolution ocean wind speed and direction database over coastal seas of Australia, New Zealand, Western Pacific islands, and the Maritime continent. It is obtained from Europe's Copernicus Sentinel-1 A and B SAR satellites from 2017 up till present. The data set is a first of its kind in the region and captures the spatial variability of
coastal ocean winds over a wide swath (250 km).

## 3 Conclusions

This chapter about air-sea fluxes has provided some information about the diverse range of air-sea flux datasets that are now available for the community to use as air-sea forcing in OOFS. Generally speaking, the datasets are defined by their spatial and temporal resolutions and are limited by associated biases which should be taken into account when choosing a dataset as
surface forcing in an OOFS. Consequently, air-sea flux data sets for OOFS should be chosen with the applications and users of the outputs in mind.



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

**Competing interests**

The contact author has declared that none of the authors has any competing interests.

**Data and/or code availability**

No data and/or code have been created as part of this manuscript

**Authors contribution**

Andreas Schiller prepared the manuscript with contributions from all co-authors.

**Acknowledgements**

The authors would like to thank the Compilation Team at the OceanPrediction Decade Collaborative Centre for their guidance and support during the drafting of this manuscript.