# Peer review of "Atmospheric Forcing as a driver for Ocean Forecasting"

_State of the Planet, 2024_

## Author Comment (AC1)

Reviewer #1:

Thank you for your constructive comments and helpful suggestions. We drafted some substantial revisions to the paper based on them – as explained below.

The report of Schiller et al., entitled 'Atmospheric Forcing as a driver for Ocean Forecasting' illustrates different ways of provisioning surface momentum, heat, and freshwater fluxes to operational ocean models.

The report presents two kinds of flux dataset sources (observational vs. numerical prediction systems) and then gives some relevant considerations about the application of atmospheric forcing to ocean forecasting systems for global vs. regional/coastal systems. The brief conclusion does not give any recommendations, except that the suitable atmospheric forcing depends on "the applications and users".

I understand that the report can not be exhaustive about the available atmospheric or flux products and can not list in details how each operational ocean forecasting system is currently driven in surface. Nevertheless, the report is here more confusing that clarifying, especially section 1.

We agree that the manuscript lacks some clarity regarding different options for forcing an OOFS at the surface. In the revised version of the manuscript we have adopted the suggestions by reviewer #1 as outlined below and substantially rewritten old section 1 (now section 2) of the manuscript.

Furthermore, to aid clarity we also added a brief introduction section.

Here are in details my main concerns:

- Using observations is, by definition, a way to drive an ocean monitoring system or to produce a (re)analysis. Obviously, using an atmospheric forecast appears mandatory to do an ocean forecast. Somehow, this is never clearly mentioned in the paper.

This important point is now included in the paper at the beginning of section 1.

- To my knowledge, surface fluxes are not directly observed by remote-sensors, but are computed using different geophysical observed variables, generally from different platforms, and using parametrization for computation. It could be interesting here to mention if there is any initiative to gather and evaluate specific satellite flux (or atmospheric near-surface parameters) products designed for operational oceanography.

We made this point about surface (heat) fluxes not being directly observed by remote sensing (old section 1.1). However, for sake of clarity we now also highlight this point at the beginning of section 2.1.

The authors of this manuscript are unaware of any dedicated initiative to gather and evaluate any satellite-based observations (or atmospheric near-surface parameters) products specifically designed for operational oceanography.

- For ocean forecasts, the use of an atmospheric forecast as surface forcing can be done by 4 methods:

- using directly the atmospheric fluxes produced by NWP systems of weather services/centres. For that, the relevant questions for OOFS are the data availability, space-time resolution and domains for regional/coastal OOFS;
- using a so-called "bulk" forcing, i.e. the near-surface atmospheric parameters. This method permits to use the ocean surface explicit variables (temperature, current, albedo) to compute inline and eventually at each time step the turbulent fluxes and the upward radiative fluxes, and so to introduce a pseudo-coupling. This method brings the same questions than the first one, plus, the choice of the surface flux parametrization that is here crucial;
- using an intermediate simplified atmospheric model (e.g. ABL1D) driven for the large-scale by the atmospheric NWP 3D fields and producing surface fluxes consistent with the ocean evolution and resolution;
- a full ocean-3D atmosphere coupling but with specific issues relative to the numerical cost and the initialisation/assimilation, but the advantages (compared to the 3 first methods) i) to have no (or for regional OOFS a lower) dependence to the data availability from external providers and ii) to ensure a two-way consistency.

We added a brief section 3 to the revised paper which addresses the above options for implementation of ocean-atmosphere fluxes into an ocean model.

In my opinion, an improved way to present information about atmospheric forcing for OOFS can be done by following the suggested outlines hereafter:

1. atmospheric forcing for ocean forecasts. There come only NWP systems as possible forcing, but with the methods and considerations explained before, and additionally the issue of open boundaries/surface forcing consistency for regional OOFS, that is well described in the current section 2.2.
2. atmospheric forcing for ocean analyses/monitoring systems. There could be a discussion of using atmospheric analyses or "observational" flux products;
3. atmospheric forcing for re-analyses/OOFS evaluation/past case studies. For this purpose, using reanalyses or any best fit of observed data is clearly recommended.

We adopted the above structure proposed by the reviewer (section 2 of the revised manuscript).

With these comments and suggestions, I recommend a revision of the paper.

---

## Author Comment (AC2)

Reviewer #2:

We thank reviewer #2 for the constructive comments and helpful suggestions. We drafted some amendments to the paper based on them – as explained below.

This paper briefly introduces the state of the art of atmospheric forcing into ocean model forecasting, with a focus on the air-sea flux datasets.

General comments:

Although I understand this is an introductory chapter, I miss a brief discussion on the main limitations of current satellite-derived flux products in terms of the spatial and temporal resolution required for ocean forcing.

To the best of our knowledge all satellite data used for calculating flux products to force today's operational ocean forecasting systems are ingested via some kind of atmospheric model, e.g. flux products as output from an NWP model or from a fully coupled ocean-atmosphere model with satellite data being assimilated. Consequently, the spatial and temporal resolution of satellite-derived flux products used in operational ocean forecasting systems strongly depends on the methods and atmospheric models applied (including their biases) to calculate these flux products. This is a complex topic for discussion which is beyond the scope of this manuscript.

Moreover, limitations of available flux products are being addressed in subsection 2.1 *Applications in Global OOFS* (4.1 in revised manuscript) and subsection 2.2 *Applications in regional and coastal OOFS* (4.2 in revised manuscript). In these subsections we discuss the limitations of global and regional flux products.

To highlight this point made by the reviewer we modified a sentence in the Conclusion section of our revied manuscript:

"This study provides some information about the diverse range of air-sea flux datasets that are now available for the community to use as air-sea forcing in OOFS. NWP systems provide the majority of flux products to force today's OOFS. Generally speaking, the quality and usefulness of these datasets are influenced by the spatial and temporal resolutions of remotely sensed and in situ observations that are assimilated into the NWP systems and are limited by associated biases which should be taken into account when choosing such datasets."

Also relevant, NWP output is not only used because of its low latency but also (and mostly) because of its ubiquity.

Agreed. We added this point to revised section 2.1 *Atmospheric forcing for ocean forecasts.*

Moreover, some more emphasis on the current limitations of the NWP output in terms of its relatively poor spatial resolution and quality in the ocean forcing context is also desirable.

As stated above, limitations of NWP outputs and the spatial resolution required are being addressed in our discussion of air-sea fluxes for global and regional models.

Regarding satellite-derived flux datasets, I believe too much (positive) attention is given to SAR-derived wind stress in the context of coastal forcing, while no explanation on the current limitations of such technique is provided. Although quite some efforts have been devoted to SAR wind retrievals over the past two decades (see publications from, e.g., Horstmann, Mouche, Grieco, Moiseev, Zecchetto, Zhu, etc.), there is currently not a single SAR wind processor that can provide a coastal wind stress product of sufficient quality and/or coverage for use in operations, while its use for OOFS development purposes must be done with caution and on a test-case basis.

On reflection, we agree with the reviewer's comment and have revised and moderated our wording at various places in the manuscript where SAR is mentioned and expanded our discussion of SAR-derived wind stresses with a cautionary note.

Specific comments:

1. L47: Wind stress is well-determined from scatterometers since SEASAT-A (1978) and ERS-1 (1991). Suggested references: Jones et al. (1982), Stoffelen and Anderson (1997), Portabella and Stoffelen (2009).

   We have updated our statement by citing two of the above publications.

2. L48-50: Similar to the CMEMS L2 OCN product, Khan et al. (2023) use a very old technique (Portabella et al., 2002) to systematically derive coastal wind vectors from SAR. Many publications (incl. Portabella et al., 2002) point out the limitations of such technique, in particular the lack of small-scale variance in the derived wind direction component (which is mostly driven by the background wind direction, i.e., the NWP wind direction). Moreover, the uncertainty in the wind direction component is then propagated into the wind speed retrieval. I would therefore not recommend the use

   As stated above, we agree with the reviewer's comment and have revised our wording at various places in the manuscript where SAR is mentioned and expanded our discussion of SAR-derived wind stresses with a cautionary note.

3. L65-66: "…has the potential to produce biases, particularly in the radiative flux fields and precipitation (Trenberth et al., 2009; Weller et al., 2022) **and in the wind stress vector components (Belmonte and Stoffelen, 2019; Trindade et al., 2020)**".

   Sentence and references have been updated accordingly.

4. L67: Please, explain why atmospheric reanalyses are suitable for OOFS development.

   The following statement has been added to the revised manuscript: "In essence, atmospheric reanalyses are often used in OOFS development and in ocean reanalyses for the following reasons: they are typically of higher quality than output from operational NWP systems (where there is less time for quality control); they are

available over an extended period of time, often covering multiple years to decades to explore various weather and climate phenomena in the ocean model in response to the atmospheric forcing; and model parameters in an atmospheric reanalysis are being kept constant over the integration period to produce a consistent data set."

5. L82-83: Please provide references. Also, briefly explain how do satellite data supplement NWP output in forcing ocean models. For example, Trindade et al. (2020) show how scatterometer-derived wind stress can be used to remove NWP model output local biases.

   Table 1 lists five different NWP and/or atmospheric reanalysis flux data sets. A reference to Table 1 has been added to this statement in the revised manuscript.

   We have adopted the suggestion by the reviewer and added the reference to Trinidade et al. (2020) as an example of how satellite data can supplement NWP output in forcing ocean models.

6. L116: Please, name a few high-resolution regional NWP models and add corresponding references.

   Statement and references added: "Examples of regional atmospheric models are the UK Met Office Unified Model–JULES Regional Atmosphere and Land configuration (Bush et al., 2023) and the Weather Research & Forecasting Model (WRF) (Skamarock et al., 2008)."

7. L125-130: Please, moderate the benefits of using SAR-derived wind (stress) for coastal OOFS development purposes.

   Agreed and moderated the benefits of using SAR-derived wind (stress) for coastal OOFS development purposes in accordance with the reviewer's comments.

References:

Belmonte Rivas, M. and Stoffelen, A.: Characterizing ERA-Interim and ERA5 surface wind biases using ASCAT, Ocean Sci., 15, 831–852, https://doi.org/10.5194/os-15-831-2019, 2019.

Jones, W. L., Schroeder, L. C., Boggs, D. H., Bracalente, E. M., Brown, R. A., Dome, G. J., … & Wentz, F. J. (1982). The SEASAT-A satellite scatterometer: the geophysical evaluation of remotely sensed wind vectors over the ocean. Journal of Geophysical Research: Oceans, 87(C5), 3297-3317. https://doi.org/10.1029/jc087ic05p03297.

Portabella, M., and Stoffelen, A., "On scatterometer ocean stress," *J. Atm. and Ocean Techn.*, **26** (2), pp. 368–382, https://doi.org/10.1175/2008JTECHO578.1, 2009.

Stoffelen, A., and Anderson, D., "Scatterometer data interpretation: derivation of the transfer function CMOD-4," *J. Geophys. Res.*, vol. 102, no. C3, pp. 5767-5780, 1997.

Trindade, A., Portabella, M., Stoffelen, A., Lin, W., and Verhoef, A., "ERAstar: a high resolution ocean forcing product", *IEEE Trans. Geosci. Rem. Sens.*, **58** (2), pp. 1337-1347, https://doi.org/10.1109/TGRS.2019.2946019, 2020.